# Polarimetric Reverberation Mapping in Medium-Band Filters

Elena Shablovinskaya [1,*], Luka Č. Popović [2,3], Roman Uklein [1], Eugene Malygin [1], Dragana Ilić [3,4], Stefano Ciroi [5], Dmitry Oparin [1], Luca Crepaldi [5], Lyuba Slavcheva-Mihova [6], Boyko Mihov [6] and Yanko Nikolov [6]

1 Special Astrophysical Observatory of RAS, 369167 Nizhny Arkhyz, Russia
2 Astronomical Observatory, Volgina 7, 11000 Belgrade, Serbia
3 Department of Astronomy, Faculty of Mathematics, University of Belgrade, Studentski trg 16, 11158 Belgrade, Serbia
4 Humboldt Research Fellow, Hamburger Sternwarte, Universitat Hamburg, Gojenbergsweg 112, D-21029 Hamburg, Germany
5 Dipartimento di Fisica e Astronomia, Università di Padova, 35122 Padova, Italy
6 Institute of Astronomy and NAO, Bulgarian Academy of Sciences, 72 Tsarigradsko Chaussee Blvd., 1784 Sofia, Bulgaria
* Correspondence: e.shablie@yandex.com

**Abstract:** Earlier, we suggested the "reload" concept of the polarimetric reverberation mapping of active galactic nuclei (AGN), proposed for the first time more than 10 years ago. We have successfully tested this approach of reverberation mapping of the broad emission line on the galaxy Mrk 6. It was shown that such an idea allows one to look at the AGN central parsec structure literally in a new light. However, the method originally assumed the use of spectropolarimetric observations, expensive in terms of telescope time, and implemented on rare large telescopes. Currently, we propose an adaptation of the polarimetric reverberation mapping of broad lines in medium-band filters following the idea of the photometric reverberation mapping, when filters are selected so that their bandwidth is oriented to the broad line and the surrounding continuum near. In this paper, we present the progress status of such monitoring conducted jointly at the Special astrophysical observatory and Asiago Cima Ekar observatory (OAPd/INAF) with support from Rozhen National Astronomical Observatory (NAO), some first results for the most frequently observed AGNs Mrk 335, Mrk 509, and Mrk 817, and the discussion of the future perspectives of the campaign.

**Keywords:** polarization; active galactic nuclei; reverberation mapping





## 1. Introduction

According to the unified model of active galactic nuclei (AGN) [1,2], the central parts of the central machine are surrounded by a gas–dust region, the so-called dusty torus. The presence of a dusty region is key to explaining the observed dichotomy of type 1 and type 2 AGN (see [3]). Characteristics of dust surrounding AGN, e.g. its location and chemical composition determine the accretion properties of the AGN. Thanks to high-angular-resolution observations of local active galaxies in the infrared (IR) (e.g., [4–8]) and molecular lines (e.g., [9,10]) now it becomes possible to obtain direct images of the dusty region, while in the optical range, this structure is still unresolvable. The development of observational capabilities made it possible to determine the geometry of the dusty region, which turned out to be different from the toroidal (see [11] for a review), and also to move from the simple models of a clumpy [12] and smooth-distributed [13] dusty "torus" to more complex ones ([7,14,15] etc.).

The equatorial scattering observed in the optical range in many central regions of type 1 AGN [16–18] is also associated with the presence of a dusty region. This process is responsible for the specific polarization signatures along the emission line profiles: an *S*-shaped swing in the polarization angle and a dip in the polarization degree along the

emission line profile [19], which cannot be explained by any other polarization mechanisms in AGN. Scattering by particles of the medium (mainly electrons) occurs in the plane of rotation of the AGN at a distance of $R_{sc}$, where the optical depth becomes greater than 1 [16,18,20]. Based on physical assumptions, $R_{sc}$ is consistent with the dust sublimation radius. In particular, refs. [21–23] use IR measurements as $R_{sc}$ for estimations of supermassive black hole (SMBH) masses by spectropolarimetric data of angle swings in broad lines. However, there are no direct observations of the equatorial scattering region, and the regions observed in IR may be located farther from the AGN center at a greater optical depth than the optical radiation scattering region.

Since the scattering and emitting regions are spatially separated, polarimetric reverberation mapping can be used to determine their sizes. The simulation done by Goosmann et al. [24] showed that the equatorially scattered polarized emission of the AGN must lag behind the continuum emission. However, the first observational test for NGC 4151 showed $R_{sc} < R_{BLR}$ [25]. Due to the use of broad-band filters covering mostly continuum, the polarization observed in NGC 4151 was contributed not only by equatorial scattering but also by sources of the polarized continuum, such as the accretion disk or the base of the jet. Shablovinskaya et al. [26] revised the approach and proposed the idea of AGN reverberation mapping in polarized broad lines. When using a polarized flux in an emission line with a continuum flux subtracted from it, the influence of other polarization mechanisms is minimized, which allows us to measure the time delay that occurs precisely due to scattering by the equatorial region. The new approach was applied to the analysis of data from spectropolarimetric monitoring of the Seyfert galaxy Mrk 6. The detected delay between the polarized emission in the broad H$\alpha$ line and the continuum at a wavelength of 5100 Å was about 100 days, which is close to the theoretical value ($\sim$115 days, [27]), but about two times less than the expected delay according to the spatial estimate of the size of the dust region, obtained by IR-interferometry ($\sim$214 days, [5]). This discrepancy requires a detailed analysis, but without static reinforcement based on monitoring other galaxies, it cannot clarify the physics of AGNs.

Spectropolarimetric monitoring, which initially underlay the broad line polarimetric reverberation method, requires not only a large amount of time on a large telescope but also the use of a device equipped with this mode, which is implemented only in a few observatories. Small telescopes are best suited for monitoring a large sample of objects. To adapt the technique for small instruments, it was necessary to switch from spectroscopy to direct images, as was done in the case of photometric reverberation mapping [28][1]. Similarly, AGN reverberation mapping in polarized broad lines at small telescopes can be implemented using image-polarimetry in mid-band filters oriented to the emission line and continuum nearby.

In this paper, we consider the adaptation of the method of polarimetric reverberation mapping of broad lines to observations with small telescopes and some preliminary results. The paper structure is as follows. Section 2 describes the observational technique used on 1- and 2-m class telescopes and the sample of the AGNs chosen for the first stage of the monitoring project. In Section 3, the first results for the three most frequently observed AGNs—Mrk 335, Mrk 509, and Mrk 817—are given, which are then discussed and compared with other estimations in Section 4. The perspectives of the observational approach are described in Section 5. The summary of the current project state is in Section 6.

## 2. Observational Technique and Sample

Since the beginning of 2020, we have been conducting polarimetric monitoring of a sample of type 1 AGN with equatorial scattering at the 1-m telescope Zeiss-1000 [30] of the Special Astrophysical Observatory of the Russian Academy of Sciences (SAO RAS), at the Copernico 1.82-m telescope of the Asiago-Cima Ekar Observatory and at 2-m telescope of Rozhen National Astronomical Observatory (NAO).

On the Zeiss-1000 telescope of the SAO RAS at different times, we used two instruments "StoP" and "MAGIC" using medium-band 250 Å-wide filters from the SED-set[2]

and 100 Å-wide filters called Sy671 and Sy685. In 2020, observations were made on the photometer-polarimeter "StoP" [31]. Using a double Wollaston prism [32] as a polarization analyzer, the device made it possible to simultaneously registered four images in the detector plane corresponding to electric vector oscillations in the directions 0°, 45°, 90° and 135° and, consequently, three Stokes parameters *I*, *Q*, and *U* within one exposure. This method of observation makes it possible to minimize the effect of atmospheric depolarization and increase the accuracy of polarimetric observations (for more details, see [33]). In the polarimetry mode of the "StoP" device with a CCD system (2k × 2k px) Andor iKon-L 936 (a detailed study of the detector is described in [34]) in the 2 × 2 binning mode the scale is $0''.42/\text{pix}$ with the field of view (FoV) for each direction of polarization $0'.9 \times 6'.1$.

Since the end of 2020, we have switched to a new device—the "MAGIC" multimode focal reducer [34,35]. Retaining all the advantages of the predecessor instrument in the technique of polarimetric observations (the linear polarization measurement accuracy is up to 0.1% for stellar sources up to 16 mag.), the new device has a large FoV: a Wollaston quadrupole prism [36], used as polarization analyzer, projects onto the CCD-detector 4 images of the input mask, corresponding to the directions of oscillation of the electric vector 0°, 90°, 45° and 135°, each with a size of $6'.5 \times 6'.5$. This allows using several local standard stars in the field of an object in differential polarimetry. When using the focal reducer with the same Andor iKon-L 936 CCD system in the 1 × 1 binning mode, the scale was $0''.45/\text{pix}$.

At Asiago Cima Ekar observatory (OAPd/INAF), data were obtained using H$\alpha$, 671 and 680 filters (70 Å-, 100 Å-, and 100 Å-widths, respectively, and centred at 656, 671, and 680 nm, respectively) and a double Wollaston prism as a polarization analyzer, placed inside the Asiago Faint Object Spectrographic Camera[3] (AFOSC) of the 1.82-m Copernico telescope. The same technique allows simultaneously obtaining four images of the input mask in different directions of polarization in the FoV $0'.8 \times 9'.4$ with Andor iKon-L 936 CCD system with a scale of $0''.51/\text{pix}$ in 2 × 2 binning mode.

The data at Rozhen National Astronomical Observatory (NAO) were obtained at the 2-m Ritchey–Chrétien–Coudé (RCC) telescope using the two-channel Focal Reducer Rozhen (FoReRo-2) [37], equipped with a double Wollaston prism and a 2k × 2k px Andor iKon-L CCD camera. The images were taken through IF642 and IF672 narrow-band filters (with 26 Å and 33 Å FWHMs, centered at 6416 Å and 6719 Å, respectively). The four images have FoV of $50''.7 \times 50''.7$ each and a scale of $0''.994/\text{px}$ in the 2 × 2 binning mode used. However, due to the infrequent observations and the small FoV not allowing one to apply the reduction technique shown below, we have not used further this data for the time series analysis.

During each observational night, we received calibration images (flat frames for each filter, bias) to correct data for additive and multiplicative errors. For each object, the series of images (at least 7 frames in each filter) were taken, the exposure times depend on the object brightness, and weather conditions and are usually ranged from 2 to 5 min. To correct statistics each frame is processed independently, and statistical evaluation is made by averaging the random value by robust methods giving its unbiased estimate. In this case, the polarimetric errors are the standard deviation of the robust distribution.

AGN observations were accompanied by observations of polarized standard stars and stars with zero polarization. Introducing the instrumental parameters $K_Q$ and $K_U$, which characterize the transmission of polarization channels, determined from observations of unpolarized standard stars, as well as $I_0$, $I_{45}$, $I_{90}$, and $I_{135}$ as the intensity at four polarization directions, we can measure three Stokes parameters:

$$I = I_0 + I_{90}K_Q + I_{45} + I_{135}K_U \tag{1}$$

$$Q = \frac{I_0 - I_{90}K_Q}{I_0 + I_{90}K_Q} \tag{2}$$

$$U = \frac{I_{45} - I_{135}K_U}{I_{45} + I_{135}K_U} \tag{3}$$

Here and below, we use $Q$ and $U$ to denote the normalized Stokes parameters.

Then, the degree of linear polarization $P$ and the polarization angle $\varphi$ as:

$$P = \sqrt{Q^2 + U^2} \tag{4}$$

$$\varphi = \frac{1}{2} \arctan\left(\frac{U}{Q}\right) \tag{5}$$

The observation technique and data reduction are described in more detail in [33]. Note here that the interstellar medium (ISM) polarization is corrected using only one local standard star in the field of the AGN, which may introduce a slight bias in measured polarization parameters. Yet, this bias is about to be small and stable within the monitoring campaign.

When the signal-to-noise ratio of the measured polarization in AGNs was small ($\sigma_P/P \gtrsim 0.7$, where $\sigma_P$ is the error of the polarization degree $P$ measurement), the polarization degree was corrected for the polarization bias [38]:

$$P_{\text{unbiased}} = P \cdot \sqrt{1 - (1.41 \cdot \sigma_P/P)^2}. \tag{6}$$

However, >95% of obtained data is of high signal-to-noise ratio ($\sigma_P/P < 0.7$).

Over the past two years, we have concentrated on observations of the 6 brightest (12–15 mag) objects in the sample (see Table 1) with equatorial scattering, confirmed by spectropolarimetric observations at the 6-m BTA SAO RAS. All type 1 AGNs are observed sequentially in the polarimetry mode in several mid-band filters, the passbands of which are spectrally oriented toward the emission of the broad H$\alpha$ line and the continuum near the line. Note that in all cases, it is the broad H$\alpha$ line that we observe since the equatorial scattering effect is most detectable there. The selection of filters for three objects from the sample Mrk 335, Mrk 509, and Mrk 817 is shown in Figures 1 and 2. Depending on the available filter sets one filter or the combination of two filters was used for obtaining the emission line flux. Observations were carried out approximately once a month, depending on the weather conditions and according to the allocated telescope time for the implementation of programs.

**Table 1.** AGN sample and calculated expected values of the polarization parameters ($Q$, $U$, $P$ and $\varphi$) in the filters used in the observations based on the data from [22]. It is important to note that the earlier published data for the objects Mrk 335 and Mrk 79 have been corrected after more thorough processing.

| Object | Filters | $Q$, % | $U$, % | $P$, % | $\varphi$, ° |
|---|---|---|---|---|---|
| Mrk 335 | SED675 | 0.28 | −0.16 | 0.32 | 165.1 |
|  | SED650 | 0.55 | −0.51 | 0.75 | 158.6 |
|  | 680 | 0.41 | −0.14 | 0.43 | 170.6 |
|  | 671 | 0.12 | −0.13 | 0.18 | 156.4 |
|  | H$\alpha$ | 0.40 | −0.34 | 0.52 | 159.8 |
| Mrk 817 | SED625 | −0.69 | −0.43 | 0.81 | 106.0 |
|  | Sy685 | −0.01 | −0.62 | 0.62 | 134.5 |
|  | Sy671 | −0.82 | 0.36 | 0.89 | 78.1 |

**Table 1.** *Cont.*

| Object | Filters | *Q*, % | *U*, % | *P*, % | *φ*, ° |
|---|---|---|---|---|---|
| Mrk 6 | SED675 | 0.16 | −0.66 | 0.68 | 141.8 |
| | SED650 | 0.44 | −0.62 | 0.76 | 152.7 |
| | SED625 | 0.33 | −0.67 | 0.75 | 148.1 |
| Mrk 79 | SED675 | −0.42 | 0.02 | 0.42 | 88.6 |
| | SED650 | −0.40 | 0.04 | 0.40 | 87.1 |
| NGC 4151 | SED650 | −0.13 | 0.18 | 0.22 | 62.9 |
| | SED600 | −0.18 | 0.24 | 0.30 | 63.4 |
| Mrk 509 | SED675 | 0.74 | −0.63 | 0.97 | 159.8 |
| | SED650 | 0.63 | −0.60 | 0.87 | 158.2 |

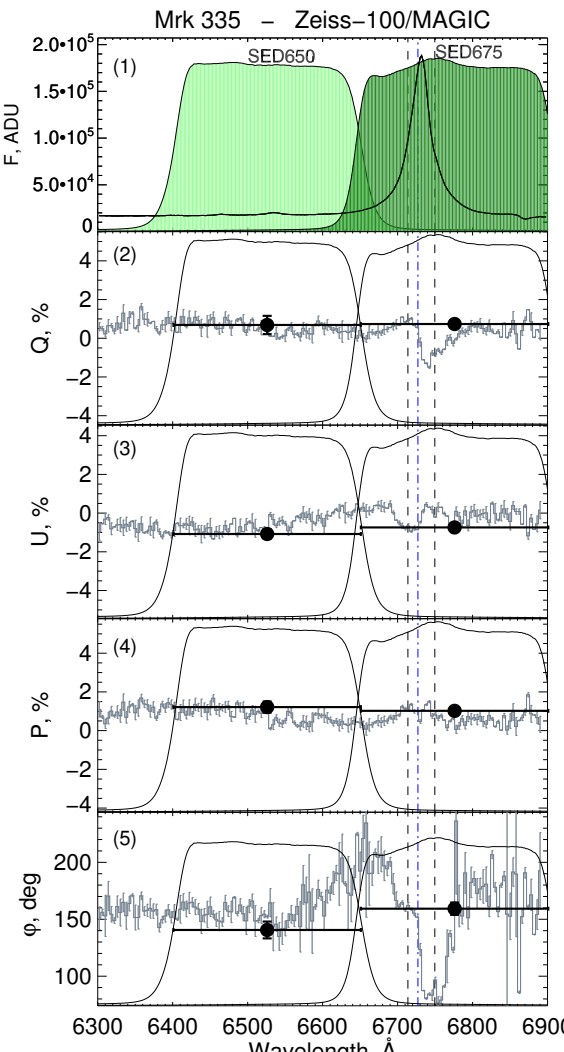 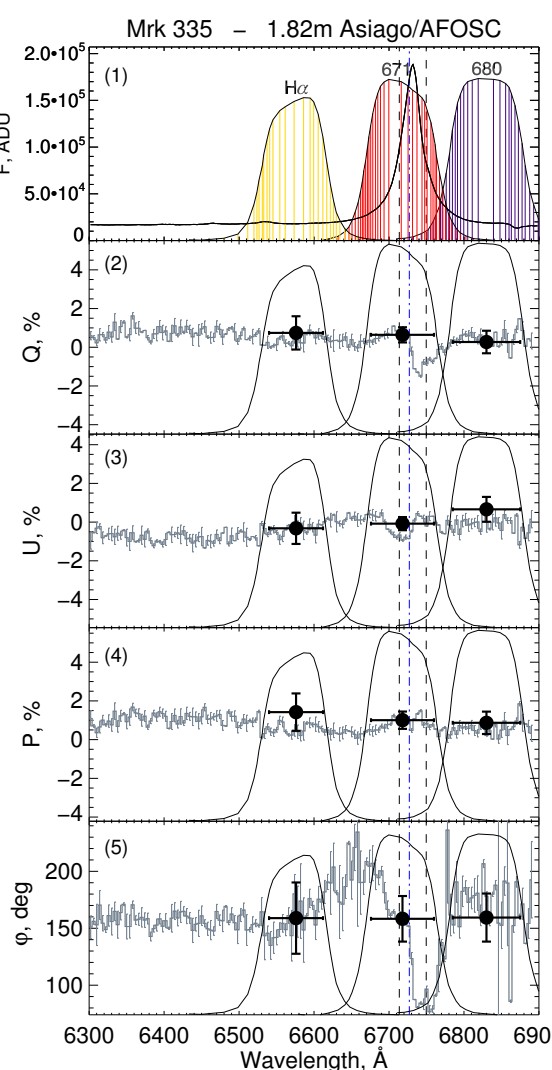

**Figure 1.** The spectropolarimetric data for Mrk 335 from [22] taken at 9 November 2013 with the overplotted selected filters. **Left:** SED675 (dark green) and SED650 (light green) transmission curves are given. In panels 2–5, the data of image-polarimetry obtained with Zeiss-1000/MAGIC 1 November 2022 are overplotted with black dots. **Right:** Hα (yellow), 671 (red), and 680 (purple) transmission curves are given. In panels 2–5, the data of image-polarimetry obtained with AFOSC 31 October 2022 are overplotted with black dots. In both figures: flux in ADU (1), the Stokes parameters *Q* (2) and *U* (3) in %, the polarization degree *P* in % (4), the polarization angle *φ* in degrees (5).

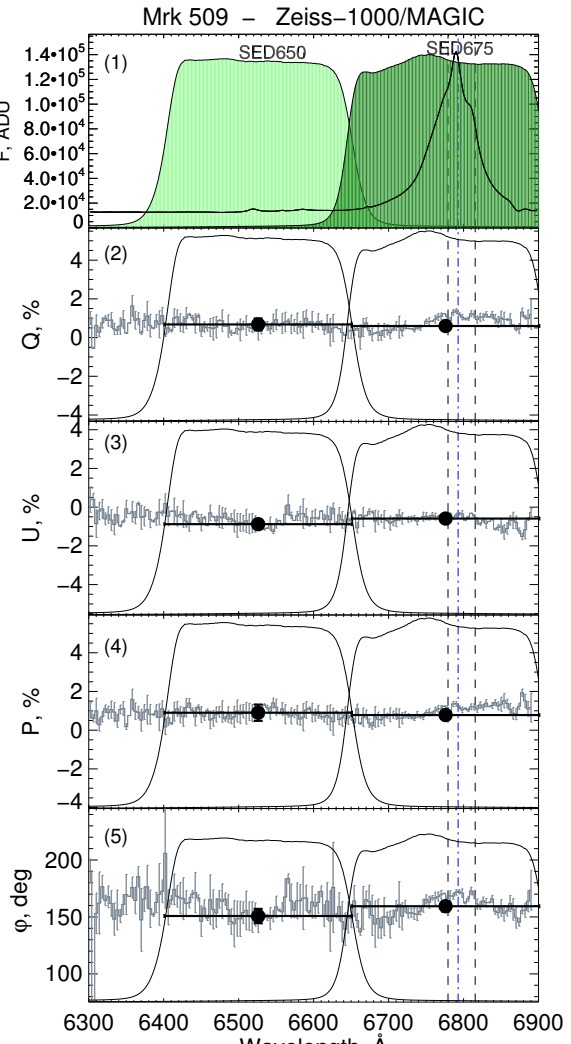

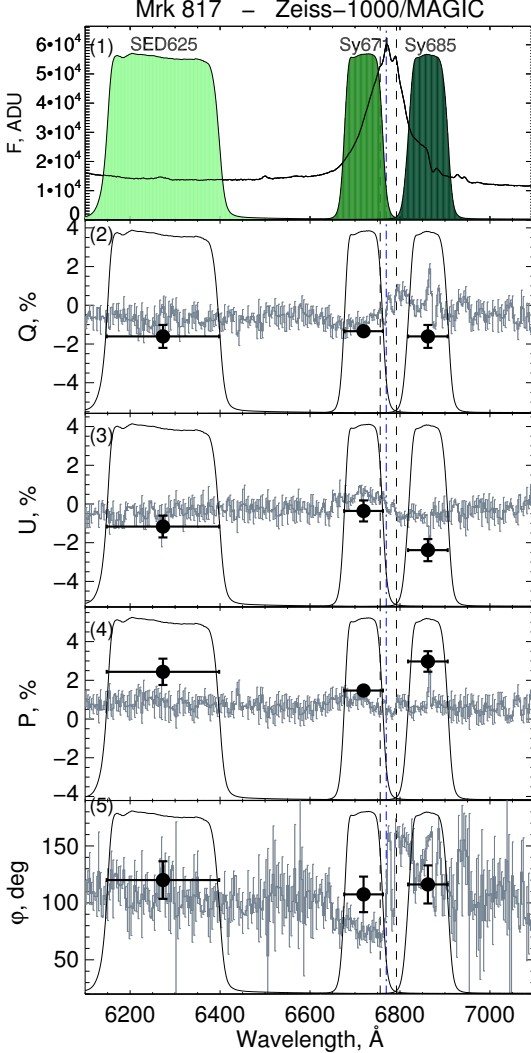

**Figure 2.** The spectropolarimetric data for Mrk 509 (**left**) and Mrk 817 (**right**) from [22] taken at 21 October 2014 and 29 May 2014, respectively, with the overplotted selected filters. **Left:** SED675 (dark green) and SED650 (light green) transmission curves are given for Mrk 509. In panels 2–5, the data of image-polarimetry obtained with Zeiss-1000/MAGIC 29 August 2021 are overplotted with black dots. **Right:** Sy685 (dark green), Sy671 (medium green) and SED625 (light green) transmission curves are given for Mrk 817. In panels, 2–5 the data of image-polarimetry obtained with Zeiss-1000/MAGIC 28 August 2021 are overplotted with black dots. In both figures: flux in ADU (1), the Stokes parameters *Q* (2) and *U* (3) in %, the polarization degree *P* in % (4), the polarization angle *φ* in degrees (5).

We estimated the expected values of the polarization effect due to equatorial scattering in the observations of all studied AGNs in medium-band filters for a broad line and continuum based on the previously obtained spectropolarimetric data [22]. Since the transmittance of medium-band filters is measured in the laboratory, we denote it as a filter's response function $filter(\nu)$ and multiply it by the spectral distribution of the polarization parameters $\xi_\nu$ [here we used $Q(\nu)$ and $U(\nu)$ in per cent] over the frequencies of the investigated AGNs, to determine its expected values $X$ (in terms of $Q$ and $U$) in specific filters:[4]

$$X = \frac{\int \xi_\nu \cdot filter(\nu) \cdot d\nu}{\int filter(\nu) \cdot d\nu} \tag{7}$$

The estimated values of $Q$ and $U$ are given in Table 1. The values of $P$ and $\varphi$ are calculated using Equations (4) and (5). It is interesting to note that in the case when the

observations are carried out in two mid-band filters, one of which is oriented to the continuum, and the second is so that the flux from the broad emission Hα line falls into it, the difference between the normalized Stokes parameters between the continuum and the line is small and does not exceed ∼0.3%, which is comparable to the linear polarization measurement error for AGN (0.1−0.2% in good weather conditions). The difference in the degree of polarization in the two filters is ∼0.1–0.4%, and the difference in the polarization angle is no more than 10 degrees. Thus, the swing seen in the spectropolarimetric observations could not be resolved by photometric polarimetry in filters, but this could indicate a difference between the emission line and continuum polarization parameters. Here, the configuration for the Mrk 817 object deserves special attention, when a broad emission line is observed in two filters oriented to the "blue" and "red" wings of its profile. For Mrk 817 (the spectrum of the object with overplotted transmission curves of the filters used is shown in Figure 2 on the right), the difference between the normalized Stokes parameters for the continuum and the line wings reaches ∼0.7%, and the deviation of the polarization angle from its value in the continuum is ±28°. It should be noted here that the Sy685 filter is also oriented to the atmospheric absorption B-band $\lambda$ = 6860–6917 Å (and Table 1 shows calculations without correction for this band). Nevertheless, the Mrk 817 case most clearly shows that using medium-band filters oriented to different wings of the Hα broad line profile, we can trace the characteristic changes in the polarization angle profile, the wavelength dependence of which acquires a characteristic *S*-shaped profile during equatorial scattering on a gas–dust torus.

## 3. First Results

We performed polarimetric observations of the AGN sample on the 1-m and 1.82-m telescopes in 2020–2022. The weather and the time allocated within the schedules did not allow us to observe objects with a high cadence, and the total amount of data on the light curves does not exceed 25 epochs, even in the case of the most regularly observed objects. Such a meagre amount of data does not allow us to get a reliable result yet. In this section, we will consider the current status of monitoring of three objects—Mrk 335, Mrk 509, and Mrk 817. The monitoring period and the number of epochs are mean values of the nonpolarized continuum, and the broad line flux and mean polarization degree of the broad line are given in Table 2. There also, we provide the measure of variability calculated using Equation (3) from [39]. The full observational data are given in Appendix A.

**Table 2.** Mean values of nonpolarized continuum and broad line flux and mean polarization degree of the broad line obtained for Mrk 335, Mrk 509, and Mrk 817 during the observations: (1) object name, (2) monitoring period (dd/mm/yyyy), (3) the number of epochs, (4) mean continuum flux in mJy, (5) variability measure of the continuum flux, (6) mean broad Hα line flux with the continuum subtracted, in mJy, (7) variability measure of the broad line flux, (8) mean polarization degree of the broad line in %. For Mrk 817, the upper values of $I_{\text{line}}$, $F_{\text{var}}^{\text{line}}$ and $P_{\text{line}}$ are for the "blue" line profile wing and the bottom values are for the "red" wing.

| (1) | Period (2) | N (3) | $I_{\text{cont}}$, mJy (4) | $F_{\text{var}}^{\text{cont}}$ (5) | $I_{\text{line}}$, mJy (6) | $F_{\text{var}}^{\text{line}}$ (7) | $P_{\text{line}}$, % (8) |
|---|---|---|---|---|---|---|---|
| Mrk 335 | 9 September 2020–1 November 2022 | 23 | 90.2 ± 7.9 | 0.101 | 140.1 ± 9.9 | 0.085 | 0.9 ± 0.3 |
| Mrk 509 | 26 May 2020–29 August 2021 | 11 | 12.2 ± 0.5 | 0.018 | 21.0 ± 2.2 | 0.092 | 1.9 ± 0.8 |
| Mrk 817 | 14 December 2020–28 August 2021 | 8 | 23.7 ± 1.1 | 0.036 | 36.9 ± 1.3 | 0.044 | 1.9 ± 0.4 |
| | | | | | 23.3 ± 1.2 | 0.072 | 2.1 ± 0.8 |

### 3.1. Mrk 335

Mrk 335 ($z = 0.025$, RA 00 06 19.5 Dec +20 12 10.6 J2000) is a well-known narrow-line Sy 1 galaxy. The signs of the equatorial scattering in broad lines were observed in Mrk 335 spectropolarimetric data for the first time in [40]. The violent polarization angle swing along the Hα line profile was confirmed in [22]; here, we present the same data obtained

at 6-m BTA telescope of SAO RAS in Figure 1. As it could be seen the polarization angle variations are of about ± 50°, yet the polarization degree changes relative to the continuum are minor. In Figure 1, left, the data of image-polarimetry obtained with Zeiss-1000/MAGIC 1 November 2022 are overplotted; in Figure 1, right, the data of image-polarimetry obtained with AFOSC 31 October 2022 at Asiago observatory are given. Note here that in all the cases, only slight differences of the polarization parameters between continuum and broad line bands are detected.

In total, 10 epochs of Mrk 335 polarimetric data were obtained with MAGIC and 13 epochs with AFOSC. Unfortunately, due to the different brightness of the field stars in the filters used in MAGIC and AFOSC, we were unable to use the same local standard. For the MAGIC data reduction, we used the reference star [GKG2008] 5 nearby at a distance of ∼1′.3 from the source, and the star TYC 1184-771-1 at a distance of ∼2′.5 for AFOSC data. The light curves polarized and broad integral line fluxes and continuum flux are shown in Figure 3. For AFOSC data, the broad line flux is the sum of fluxes measured in two filters (671 and 680). In all cases, the fluxes are given in mJy. One can see that despite the broad line curves the continuum flux light curves seem to behave in a different way in the data sets obtained with MAGIC and AFOSC. That was the reason not to merge the light curves not to introduce the systematical errors in the correlation analysis.

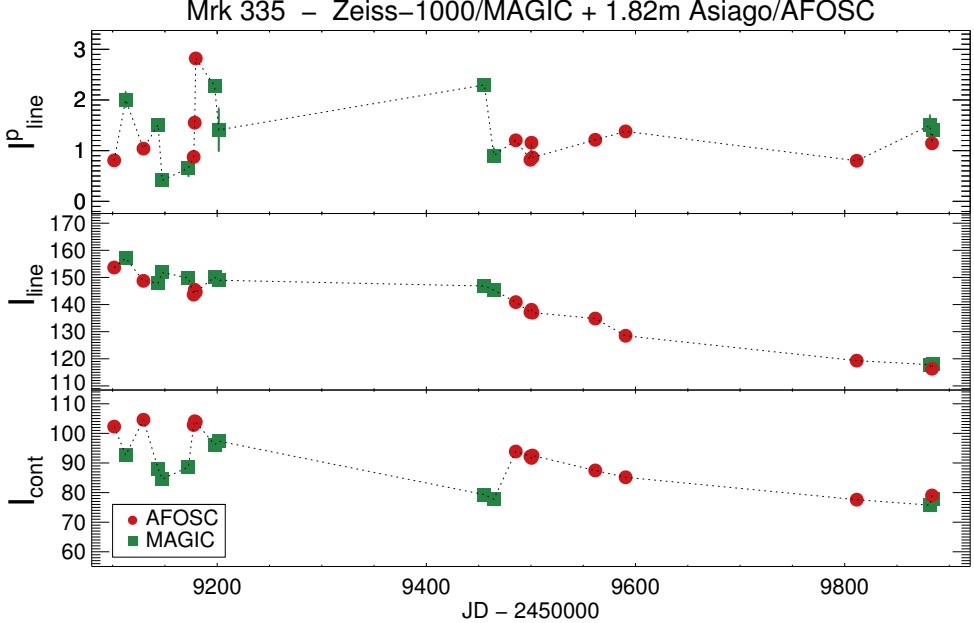

**Figure 3.** Mrk 335 light curves. From top to bottom: polarized broad line flux $I_{line}^p$, integral broad line flux $I_{line}$ with subtracted continuum and integral continuum flux $I_{cont}$. Fluxes are given in mJy. Red circles are used to denote the AFOSC data, and green squares are for the MAGIC data. For AFOSC, the broad line flux is the sum of fluxes measured in two filters (671 and 680).

Additionally, we have estimated the polarization of Mrk 335 using observations at Rozhen observatory taken at 15 August 21. For IF642 oriented to the continuum $P = 1.25 ± 0.26\%$, and $\varphi = 78°.5 ± 6°.0$; for IF672 oriented to the Hα emission line $P = 1.13 ± 0.13\%$, and $\varphi = 89°.0 ± 3°.3$. Here, one can detect the slight difference of the absolute polarization level, particularly seen in the polarization angle. However, due to the lack of the local standard stars in the FoV of the source we are not able to correctly take into account the atmospheric depolarization, ISM effects, etc. Moreover, the flux calibration is also absent which makes it complicated to exclude the polarized continuum flux from the polarized line flux in a proper way. Thus, these data illustrate the possibilities of the medium-band polarimetry at 2-m telescope, yet it is not used for the further analysis.

To determine the time delay between the light curves, we performed a cross-correlation analysis using two approaches. As the main analysis tool, we used the JAVELIN code [41–43], widely used in AGN reverberation mapping campaigns. In Figure 4 the results of the JAVELIN analysis of the time delay of the polarized broad line emission $I_{line}^{p}$ relative to the variable continuum flux $I_{cont}$ for AFOSC (red histogram) and MAGIC (green histogram) data are presented. Additionally, we conducted a joint analysis of both light curves, combining them so that one of the light curves is shifted in time relative to the other by more than the duration of the entire monitoring period. The results of the analysis of this synthetic curve are shown in Figure 4 in black and provide only additional information. Similarly, in Figure 5 the time delay of nonpolarized broad line emission $I_{line}$ relative to the continuum $I_{cont}$ is analyzed. Also, to estimate the delay between the light curves, we used the code ZDCF [44,45]. Separately for the MAGIC and AFOSC light curves, cross-correlation analysis using ZDCF did not show results due to large uncertainties caused by a small number of points. The results of estimating the delay between the combined synthetic curves are given in Figures 4 and 5 in grey.

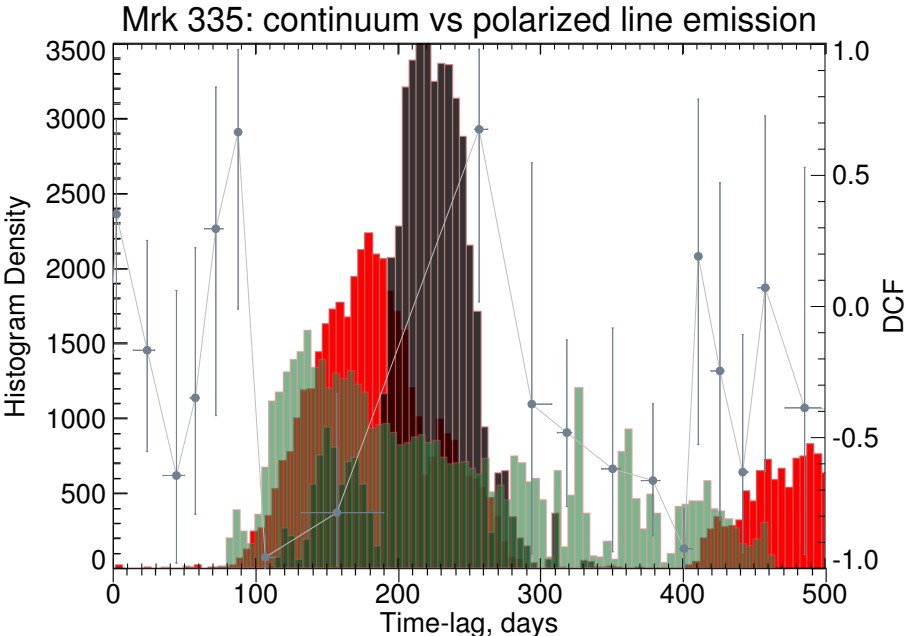

**Figure 4.** Time delay analysis of the polarized broad line emission $I_{line}^{p}$ relative to the variable continuum flux $I_{cont}$ for Mrk 335. Histograms show the results of JAVELIN analysis based on AFOSC data (red histogram), MAGIC data (green histogram), and combined time-shifted data (black histogram). On the left *y*-axis the frequency of occurrence of parameter values sets during MCMC sampling is shown. 5000 sets of parameter values were used in the simulation. The grey curve shows the results of the ZDCF analysis of the combined time-shifted data (values are given on the *y*-axis on the right).

Despite the number of epochs comparable to what we previously obtained for Mrk 6 in spectropolarimetric mode [26], the analysis of the delay between $I_{line}^{p}$ and $I_{cont}$ does not show an unambiguous peak for Mrk 335. In Figure 4 it can be seen that the histogram of estimates of time delays for AFOSC and MAGIC data is close, about 180 and 150 days, respectively, but the peak of the time-lag distribution has an error of 25–40%. Synthetic data show two peaks at $224 \pm 24$ days and $157 \pm 18$ days, where the given errors are formally calculated as the standard deviations of the given Gaussian-like peaks. Here, the larger peak is definitely an artifact, since it is repeated in the analysis of photometric data (Figure 5). The second peak is ~4 times weaker than the first one, but its position roughly coincides with other estimates. Thus, we see the tendency of the $I_{line}^{p}$ light curves to show a

delay of about 150–180 days. However, such a time lag is close to the half of year, which characterize the typical length of the observational periods of the source, and is shorter than the gaps between these periods. This might indicate the measured value as the analysis artefact. Additionally, we performed data analysis of the time delay of $I_{\text{line}}$ relatively to $I_{\text{cont}}$ to estimate $R_{\text{BLR}}$ if possible. JAVELIN histograms for AFOSC and MAGIC data, as well as analysis of synthetic light curves by the ZDCF method, indicate an estimated delay between $73 \pm 18$ and $87 \pm 17$ days. The AFOSC data separately demonstrate a peak at the value of $27 \pm 17$ days, which is close to the cadence of observations (about 1 time per month) and can be an artifact of analysis.

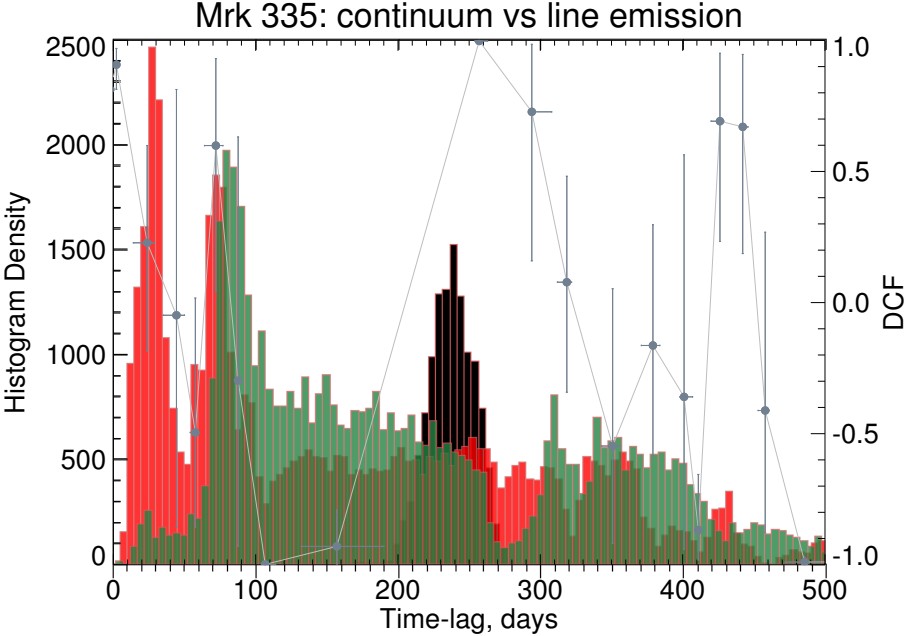

**Figure 5.** Time delay analysis of the broad line emission $I_{\text{line}}$ relative to the variable continuum flux $I_{\text{cont}}$ for Mrk 335. The coloured histograms and labels are the same as in Figure 4.

### 3.2. Mrk 509

As well as in the case of Mrk 335, Mrk 509 ($z = 0.035$, RA 20 44 09.8 Dec $-10$ 43 24.7 J2000) was observed in spectropolarimetric mode firstly in [40] and later in [22]. The latter data were used in Figure 2 (left). As can be seen in the figure, we selected for observations two medium-band (FWHM = 250 Å) filters oriented to a broad line and a continuum near. The overplotted image-polarimetry data was obtained with Zeiss-1000/MAGIC 29 August 2021. As in the case of Mrk 335, Mrk 509 shows only a slight difference in the polarization parameters between continuum and broad line bands, more detectable in the polarization angle variations.

As far as the object can be observed for only four months a year, in 2020–2021, we gained only 11 epochs using Zeiss-1000/MAGIC. To reduce the data, we used the reference star TYC 5760-1396-1 nearby at a distance of $\sim 1'.5$ from the source. The light curves are shown in Figure 6. For all measured fluxes $I_{\text{line}}^{p}$, $I_{\text{line}}$, and $I_{\text{cont}}$ the variability is observed, and the pattern of $I_{\text{line}}^{p}$ variations differs from other light curves. The curves show a large gap between the observational epochs associated with the inability to observe the object evenly throughout the year.

To estimate the time-delay in a broad polarized line, we applied the JAVELIN code to the received data. It turned out that despite a small number of epochs, the analysis revealed an unambiguous peak at $114_{-8.8}^{+12.7}$ days (Figure 7). We have also applied the JAVELIN analysis to the data taken only in 2020 excluding the epochs from 2021, and we have obtained the same time-delay. This estimate corresponds to the size of the dusty region

∼0.1 pc. Note, however, that the ZDCF analysis did not show a significant correlation. Additionally, we performed 2020 data analysis of the time delay of $I_{\text{line}}$ relatively to $I_{\text{cont}}$ to estimate $R_{\text{BLR}}$ following the Mrk 335 case. JAVELIN histogram is shown in Figure 8 demonstrating two clear peaks at $39 \pm 5$ days and at $85 \pm 11$ days (which is approximately $39 \pm 5$ days $\times$ 2). Taking into account that the median cadence of observations in 2020 is ∼16 days, we could not unambiguously make a conclusion about the origin of the double-peaked correlation histogram.

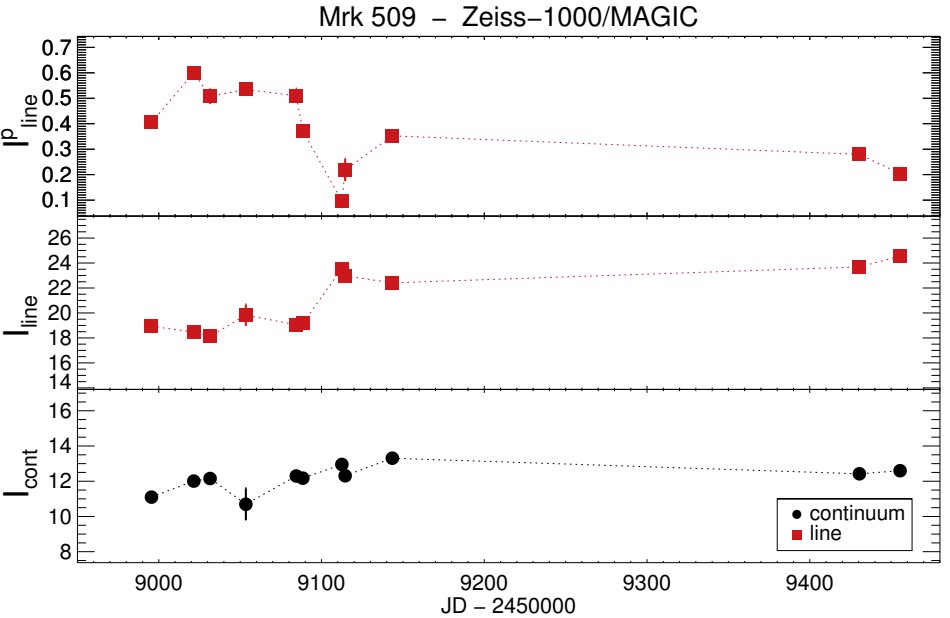

**Figure 6.** Mrk 509 light curves obtained with MAGIC. From top to bottom: polarized broad line flux $I_{\text{line}}^{p}$ and integral broad line flux $I_{\text{line}}$ with subtracted continuum and integral continuum flux $I_{\text{cont}}$. Fluxes are given in mJy.

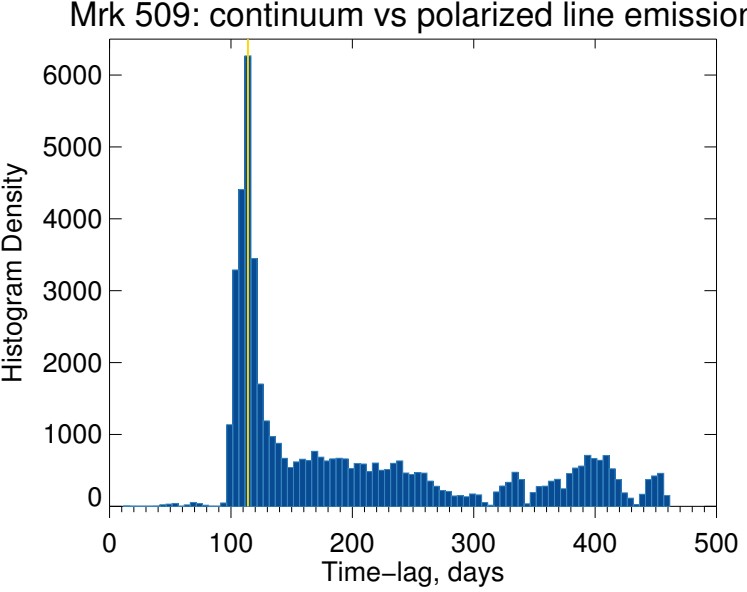

**Figure 7.** Time delay analysis of the polarized broad line emission $I_{\text{line}}^{p}$ relative to the variable continuum flux $I_{\text{cont}}$ for Mrk 509. Histogram show the results of JAVELIN analysis based on MAGIC data. On $y$-axis the frequency of occurrence of parameter values sets of during MCMC sampling is shown. 10,000 sets of parameter values were used in the simulation. The time-delay estimation equal to $114_{-8.8}^{+12.7}$ days is shown with the yellow vertical line.

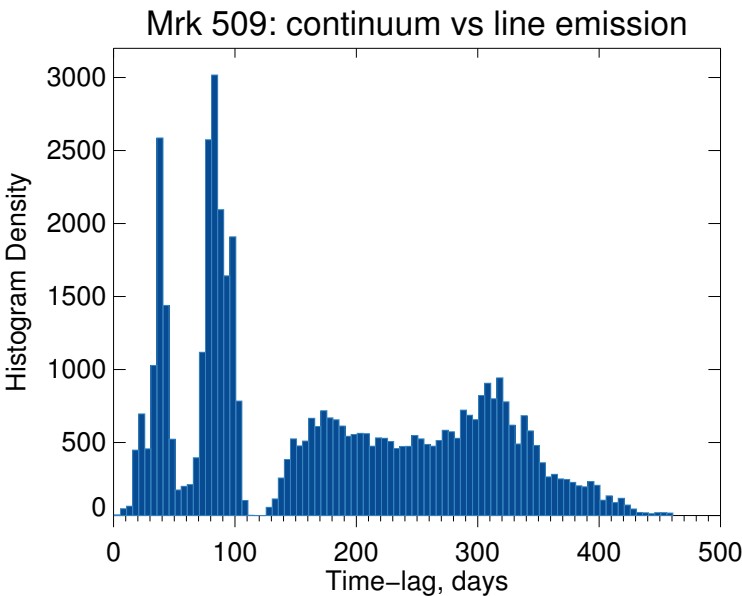

**Figure 8.** Time delay analysis of the broad line emission $I_{\text{line}}$ relative to the variable continuum flux $I_{\text{cont}}$ for Mrk 509. Histogram show the results of JAVELIN analysis based on MAGIC data. On *y*-axis the frequency of occurrence of parameter values sets of during MCMC sampling is shown. 10,000 sets of parameter values were used in the simulation.

### 3.3. Mrk 817

Mrk 817 ($z = 0.031$, RA 14 36 22.1 +58 47 39.4 J2000) is a Sy 1.2 AGN, where the equatorial scattering was discovered in [22]. The data published in that work is presented in Figure 2, right, where the transmission curves of the two filters 100 Å-width oriented to "red" and "blue" broad line wings and selected for monitoring are also shown. A broader (FWHM = 250 Å) filter was chosen for continuum polarimetry. The spectropolarimetric data demonstrate small changes in the polarization degree *P* and a violent switch of the polarization angle $\varphi$ along the line profile. In Figure 2, right, the data of Mrk 817 image-polarimetry obtained on the MAGIC 28 August 2021 device is also plotted. Comparing the values obtained in two filters oriented to different wings of the lines, one can see a significant difference in the polarization parameters. This indicates that the use of a similar filter configuration in image-polarimetry mode may be an alternative approach for identifying signs of equatorial scattering using small-class telescopes or large instruments for observations of faint AGNs where the spectropolarimetric data show too low a signal-to-noise ratio. Note here that in Figure 2 for the Mrk 817 data, there are visible differences in the polarization parameters between the observations of 2014 and 2021, especially in Sy685 band. In this case, it is important to obtain newer spectropolarimetric data in order to confirm whether such a difference is the result of the influence of external factors (e.g., the variability of atmospheric B-band 6860–6917 Å) or internal changes in the spectrum of Mrk 817 in polarized light.

During the Mrk 817 monitoring, 8 epochs of observations were obtained using the MAGIC device in the period from December 2020 to August 2021. To reduce the data, we used a reference star of comparable brightness in the field of the object (RA 14 36 06.7 +58 50 38.4 J2000) at a distance of $\sim 3'.6$ from the source. The data were obtained relatively evenly, once or twice every two months. Unfortunately, during the monitoring period, Mrk 817 did not show significant variability either in the continuum or in the broad line. The light curves of Mrk 817 are given in Figure 9. It can be seen that $I_{\text{line}}$ does not show differences between "red" and "blue" wings in integral light (see the middle panel in Figure 9). However, in polarized light, for several epochs of observations, both the difference in $I_{\text{line}}^{p}$ between filters is visible (e.g., in the epoch of 18 December 2020), and a violent change of $I_{\text{line}}^{p}$ between epochs (e.g., 2 July 2021 and 7 July 2021, see the upper panel

in Figure 9). The meagre amount of data with no significant variability did not allow us to obtain any result in cross-correlation analysis.

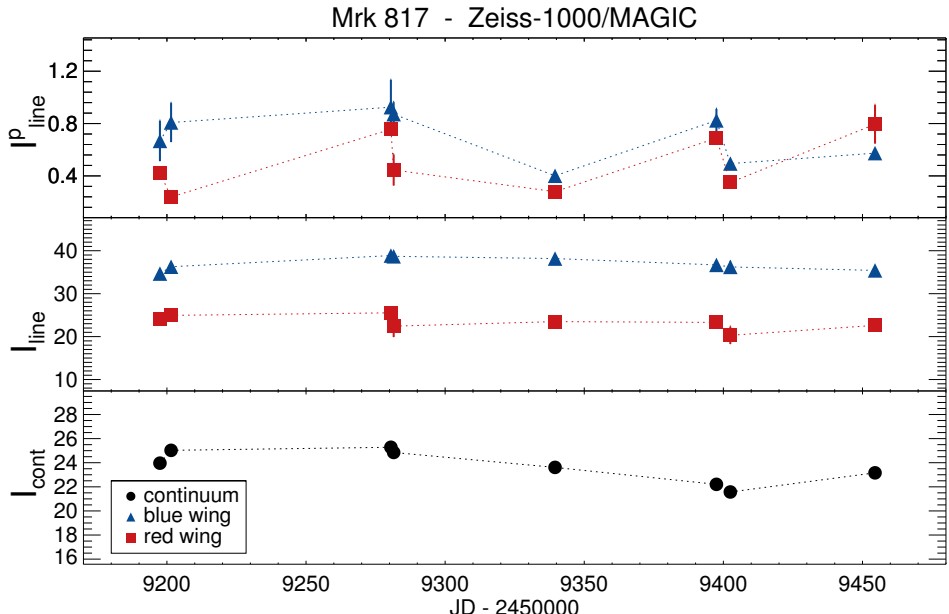

**Figure 9.** Mrk 817 light curves obtained with MAGIC. From top to bottom: polarized broad line flux $I^p_{line}$ and integral broad line flux $I_{line}$ with subtracted continuum and integral continuum flux $I_{cont}$. Red and blue dots in $I^p_{line}$ and $I_{line}$ light curves denote the fluxes of the "red" and "blue" broad line profile wings, respectively. Fluxes are given in mJy.

## 4. Discussion

Since 2020 due to the lack of stable weather meeting our requirements in two observatories involved in the project (SAO RAS and Asiago) we have not reached the desired cadence when observing a sample of objects, and the total number of epochs obtained has reached 23 for only one object. Despite this, we managed to obtain some first results for three sample objects Mrk 335, Mrk 509, and Mrk 817, presented in this paper. As these AGNs are studied in deep detail in various multi-wavelength campaigns, here we discuss our results in comparison with the measurements given in the literature to investigate if the provided estimations are reliable.

Mrk 335 was intensively studied in numerous reverberation mapping campaigns. The BLR size $R_{BLR}$ was measured as $16.4^{+5.2}_{-3.2}$ [46,47], $17.3^{+4.9}_{-4.3}$ [48], $15.7^{+3.4}_{-4.0}$ [49], $14.3 \pm 0.7$ [50,51], $10.6^{+1.7}_{-2.9}$ [52], $17.0^{+2.5}_{-3.2}$ lt days [53] in H$\beta$ broad line and $20.5^{+2.0}_{-2.8}$ lt days [28] in H$\alpha$ broad line. Given estimations of the BLR size are $\sim$10 times larger than the accretion disk size of $\sim$1 lt day [54]. According to the scale relation from [22], $R_{sc} \simeq 5.1 R_{BLR}$, so $R_{sc}$ for Mrk 335 could be estimated as $\sim$70 lt days. IR reverberation mapping in $K$ band provided $R_{IR} \approx 166$ lt days [55] which is two times greater than a value obtained using the scale relation. Lyu et al. [56] found the size of the dusty region in *WISE W1* band $R_{W1} \approx 1300$ lt days. According to the relation of the dusty region sizes in different bands $R_K:R_{W1} = 0.6:1$ given in the same paper, $R_K \approx 770$ lt days which is much larger than other estimations and seems not to be reliable. Thus, the value of the polarized emission line time lag is predicted to be in $\sim$70–170 days range. Throughout our polarimetric reverberation mapping monitoring the polarized H$\alpha$ emission showed a delay of about 150–180 days, which is in good agreement with the predictions of the size of the dusty region for Mrk 335. However, we prefer to refrain from the statement of such a result, primarily due to the fact that such an estimation may be caused by a correlation artefact since it is close to the value of 1/2 year. Moreover, as it was mentioned above, this estimate is longer than the length observation periods of the source, but shorter than the gaps between them. In addition, the

maxima of cross-correlation functions have large uncertainties of 25–40%, which makes them unconfident. Moreover, the results of the analysis of the corresponding photometric data do not coincide with the known estimates of $R_{\text{BLR}}$ for Mrk 335. These facts give reasons to doubt the sustainability of the results we have obtained so far.

Sy1 galaxy Mrk 509 was also studied in multiple campaigns covering the entire electromagnetic spectrum (see, e.g., [57] for a review). The BLR size $R_{\text{BLR}}$ was measured as $76.7^{+6.3}_{-6.0}$ [46,47] and $79.6^{+6.1}_{-5.4}$ lt days [49] in H$\beta$ broad line. From our estimations, relying on the maximum value from the double-peaked histogram in Figure 8 $R_{\text{BLR}} = 85 \pm 11$ lt days which coincides with the measurements given in the literature. However, we cannot still explain the existence of the second estimation of the time delay being two times shorter. The estimations of $R_{\text{BLR}}$ predict a very extended BLR region, which is much larger ($\sim$40 times) than the accretion disk size of $\sim$2 lt day [58,59], on the one hand, and only $\sim$2 times less then IR reverberation mapping in $K$ band estimations $R_{\text{IRRM}} \approx 131$ lt days [55]. GRAVITY Collaboration et al. [60] resolved the hot gas structure in Mrk 509 with VLTI/GRAVITY near-infrared interferometry and measured the size of the dusty region $R_{\text{IRIF}} \approx 296 \pm 30$ lt days. Using given $R_{\text{BLR}}$ measurements and the scale relation from [22], $R_{\text{sc}} \approx 408$ lt days. While $R_{\text{IRIF}} > R_{\text{IRRM}}$ is usually predicted (see [26] for references), $R_{\text{sc}}$ should be less then dusty structures in AGN. Apparently, such controversial measurements rise issues of the dusty region size. During our monitoring, we estimated $R_{\text{sc}} \approx 114^{+12.7}_{-8.8}$ lt days, or $\sim$0.1 pc. A comparison with near-IR torus interferometric data [60] shows that the equatorial scattering region is 2 times smaller than the radius of the dusty structure in the IR band, which is similar to what we previously obtained for Mrk 6 [26]. However, our estimate of $R_{\text{sc}}$, although consistent with the estimates of the size of the dusty region obtained by two independent methods, is only $\sim$1.3–1.6$R_{\text{BLR}}$. In general, all other estimates of the size of structures inside the central Mrk 509 parsec obtained independently indicate that $R_{\text{BLR}}$ is most likely overestimated, for example, due to the presence of outflows driven by AGN observed for Mrk 509 (e.g., [61]). This reveals the necessity of more intensive homogeneous monitoring in polarized and integral light.

Mrk 817 is in the focus of several vast monitoring campaigns, e.g., AGN STORM 2 [62]. The BLR size $R_{\text{BLR}}$ was measured as $15.0^{+4.2}_{-3.4}$ [46,47], $21.8^{+2.4}_{-3.0}$ [49], $14.0^{+3.4}_{-3.5}$ lt days[63] in H$\beta$ broad line, and $28.3^{+2.1}_{-1.8}$, $26.8^{+2.8}_{-2.5}$, $51.7^{+14.9}_{-1.3}$ lt days simultaneously in H$\beta$, H$\gamma$ and FeII lines, respectively [64]. Given estimations of the BLR size are $\sim$3–6 times larger than the accretion disk size of $\sim$4.5 lt day [54]. Mrk 817 was observed within IR RM monitoring and the dusty region size was estimated as $R_{\text{IRRM}} = 89 \pm 9$ lt days [55]. This coincides with the expected estimates of $R_{\text{sc}}$ using the scale relation $R_{\text{sc}} \approx 95 \pm 15$ lt days, which is predicted for our measurements. Due to the insignificant variability of the polarized and non-polarized fluxes, despite the monitoring period being two times longer than the expected time lag no result was obtained for Mrk 817. However, the variability of blue and red wings of polarized broad H$\alpha$ line is intriguing enough to go on with the observations more intensively.

## 5. Future Perspectives

The new approach of polarimetric reverberation mapping in broad lines looks promising since it can provide additional information about the size of structures in AGN, and therefore, better understand the nature of processes associated with accretion onto SMBH. As we have shown in the given paper, the technique in medium-band filters together with one-shot differential polarimetry is suitable for small telescopes, yet needs a careful adaptation. It is important to note that the polarimetry of faint polarized sources, in contrast to, e.g., differential photometry, put high restrictions on permissible weather conditions. Even weak cirruses or a haze can significantly depolarize the radiation of observed objects, and the variability of atmospheric transparency between exposures significantly degrades the quality of data. Here we consider several issues related to the adaptation of observations in the framework of monitoring.

1. Filter selection. At the beginning of the monitoring observations, we selected filters from our existing sets (see Section 2), focusing on our experience of observations in the framework of photometric reverberation mapping of AGN [65,66]. Because of this, we mainly aimed to use pairs of 250 Å-width filters oriented to a broad line and a continuum near. However, as shown in Table 1 via the convolution of spectropolarimetric data with filter transmission and in Figures 1 and 2 using the example of image-polarimetric data for Mrk 335 and Mrk 509, this strategy does not always seem optimal. This is due to that since the variations of the polarization parameters along the wavelength during equatorial scattering are small, and $\varphi$ has an *S*-shaped profile, even the 250 Å-width filter may be too broad, and the average value of the line polarization in the filter, summing up by wavelengths, will not differ markedly from the continuum. However, it is important to note that the differences are small when we consider polarization normalized by intensity. When the polarized flux in the line is considered with the subtracted polarized flux of the continuum, the behaviour of the variability in the broad line begins to differ significantly in polarized and nonpolarized light. It corresponds to the fact that we see this radiation coming from different regions of the AGN. This is what we observe in the case of objects Mrk 335 and Mrk 509. Thus, it can be argued that even if a medium-band filter covering the whole line profile is selected, the variable flux $I^{p}_{\text{line}}$ is detectable.

In cases of bright AGNs, a more optimal strategy may be to choose narrower filters oriented on different sides from the centre of a broad emission line. In our case, we were able to implement this by using 100 Å-width filters for Mrk 817 monitoring. The light curves we obtained are not yet sufficient to reveal the approach efficiency within the monitoring framework. However, as was shown above, such a strategy is more suitable to check AGNs for signs of equatorial scattering in polarized light efficiently using telescope time.

Another problem for us was the combination of observational data obtained using a different set of filters. One can see this in the example of Mrk 335 observations, which were carried out at the 1-m telescope of the SAO RAS and 1.82-m in Asiago. Having the same trend, the variability of radiation, especially in the continuum, differs significantly between the data obtained in 250 Å-width filter SED650 and in 70 Å-width H$\alpha$. It is unlikely that the reason for this difference was the AGN emission lines being on the transmission edge of the SED650 filter, e.g., the [FeX] 6374 Å line. The more likely reason may be that different reference stars were used when processing the data sets obtained from MAGIC and AFOSC. In any case, the data obtained require additional analysis.

2. Aperture selection and host-galaxy subtraction. Two objects presented in this article, Mrk 335 and Mrk 509, are almost star-shaped sources. Their host-galaxies, which fitting can be found in [49], have a small contribution to the optical band. In our observations, taking into account the image quality, the profile of objects was indistinguishable from the profiles of stars in the field. Thus, we were able to choose the size of the aperture for photometry so that the signal-to-noise ratio was maximum. However, when the host galaxy is extended, the choice of aperture is complicated, as the larger the aperture size, the more galactic flux is recorded, lowering the contrast of the polarized radiation of the nucleus. For Mrk 817 polarimetry, a fixed aperture size of $\sim 4''$ was chosen so that when processing data with different image quality (data with a seeing better than $3''$ was used), the same contribution of the host-galaxy would be inside the aperture. However, greater accuracy will be achieved if, when processing AGN images with extended galaxies, a galaxy model is subtracted from the frames. If this is not so critical in the case of Mrk 817, then for, e.g., NGC 4151, where the host-galaxy has a size $>3'$ and the contrast of the nucleus is relatively small, subtraction of the galaxy fitting may be necessary to construct the light curves. This should be the subject of a separate detailed check.

3. Cadence of observations. Currently, based on simulated AGN light curves, attempts are being made to determine the optimal cadence for reverberation mapping observations. Improving cadence leads to fewer artifacts in cross-correlation analysis, but requires a large amount of telescope time. The upper limit for time resolution is the expected time delay since when observations are made with a lower frequency, the observed variability will not

be related. For example, Kovačević et al. [67] offer ∼5-days cadence for estimates of the accretion disk size of typically 1–10 lt days size using LSST. Woo et al. [68] suggested having a factor of 5 or better time resolution for a given time lag. Due to the time allocation on the telescopes, the typical cadence of our observations was planned to be about 1 month. Such a cadence is close to optimal with the expected sizes of $R_{sc}$ for Mrk 509; for Mrk 335 and Mrk 817, a cadence of ∼20 days would be more effective. However, due to weather conditions, it turned out to be impossible to conduct observations every month, so the real-time resolution is worse. Moreover, our estimates show that it is important to simultaneously measure $R_{sc}$ and $R_{BLR}$ to improve the scale relation (which may, generally speaking, have a different appearance for different objects). In this case, observations should be carried out at least 1 (preferably 2) times a week. Such a cadence can be achieved using a telescope, observations on which are fully oriented only for such a task. According to the adaptation of the method of reverberation mapping of the polarized lines to observations on a 1-m-class telescope, such a project has prospects for development.

## 6. Conclusions

We presented the first results of reverberation mapping in polarized broad lines conducted at the 1-m telescope of SAO RAS and at 1.82-m at Astronomical observatory Asiago. Since 2020, we obtained the first results for the three most frequently observed objects from our sample of type 1 AGNs with equatorial scattering, namely, Mrk 335, Mrk 509, and Mrk 817.

- For Mrk 335, the measured dusty region size is $R_{sc} \sim 150$–180 lt days. This result coincides with the values predicted concerning the several estimations of the dusty structure in the IR band and measurements of $R_{BLR}$ via optical reverberation mapping campaigns. However, due to the irregular observations, the monitoring is going on to check whether our result is a cross-correlation artefact.
- For Mrk 509, we obtained $R_{sc} \approx 114^{+12.7}_{-8.8}$ lt days, or ∼0.1 pc. This is 2 times smaller than the radius of the dusty structure in the IR band.
- For Mrk 817, no result is obtained due to the low variability of the object during the monitoring period. However, observations of the polarized flux in the two line profile wings demonstrate a sharp variability between epochs as well as a significant difference in the polarized flux in the two wings during one epoch. This shows the potential possibility of recording the delay of a polarized signal of a broad line in different parts of its profile.

**Author Contributions:** Conceptualization, E.S. and L.Č.P.; Methodology, E.S. and L.Č.P.; Software, E.S. and R.U.; Validation, D.I.; Formal analysis, L.Č.P., R.U. and E.M.; Investigation, E.S. and E.M.; Data curation, E.S., D.I., S.C., D.O., L.C., L.S.-M., B.M. and Y.N.; Writing—original draft, E.S.; Writing—review & editing, L.Č.P., E.M. and D.I.; Visualization, E.S. and E.M.; Supervision, L.Č.P.; Project administration, E.S. All authors have read and agreed to the published version of the manuscript.

**Funding:** E.S., E.M. and R.U. were supported by RFBR grant, project number 20-02-00048 while conducting observations on 1-m telescope of SAO RAS, reducing and analyzing the polarimetric data. L.Č.P., and D.I. acknowledge funding provided by Astronomical Observatory (the contract 451-03-68/2022-14/ 200002) and by University of Belgrade-Faculty of Mathematics (the contract 451-03-68/2022-14/200104), through the grants by the Ministry of Education, Science, and Technological Development of the Republic of Serbia. L.S.M. and B.M. acknowledge the project "Reverberation mapping of quasars in polarized light" within the agreement between Bulgarian Academy of Sciences and Serbian Academy of Sciences and Arts, 2020–2022. D.I. acknowledges the support of the Alexander von Humboldt Foundation.

**Institutional Review Board Statement:** Not applicable.

**Informed Consent Statement:** Not applicable.

**Data Availability Statement:** The observational data underlying this article is available on request 1 yr after the publication of this paper.

**Acknowledgments:** Observations with the SAO RAS telescopes are supported by the Ministry of Science and Higher Education of the Russian Federation. The renovation of telescope equipment is currently provided within the national project "Science and Universities".

**Conflicts of Interest:** The authors declare no conflict of interest.

## Appendix A

**Table A1.** Observed values of nonpolarized continuum and nonpolarized and polarized broad line flux for Mrk 335, Mrk 509 and Mrk 817 : (1) date of observations (dd/mm/yyyy), (2) the same in Julian form JD-2450000, (3) continuum flux in mJy, (4) the Stokes $Q$ parameters of the continuum flux in %, (5) the Stokes $U$ parameters of the continuum flux in %, (6) broad H$\alpha$ line flux with subtracted continuum, in mJy, (7) the Stokes $Q$ parameters of the line flux with subtracted continuum, in %, (8) the Stokes $U$ parameters of the line flux with subtracted continuum, in %, (9) polarized broad H$\alpha$ line flux with subtracted continuum, in mJy. For Mrk 817, the upper values of $I_{\text{line}}$, $Q_{\text{line}}$, $U_{\text{line}}$ and $I_{\text{line}}^p$ are for the "blue" line profile wing and the bottom values are for the "red" wing. The values are calculated by robust average, and the errors are the robust standard deviation (see [69] for more details).

| Date (1) | JD (2) | $I_{\text{cont}}$ (3) | $Q_{\text{cont}}$ (4) | $U_{\text{cont}}$ (5) | $I_{\text{line}}$ (6) | $Q_{\text{line}}$ (7) | $U_{\text{line}}$ (8) | $I_{\text{line}}^p$ (9) |
|---|---|---|---|---|---|---|---|---|
| | | | | Mrk 335 (MAGIC) | | | | |
| 20 September 2020 | 9112 | $92.6 \pm 0.1$ | $0.0 \pm 0.1$ | $-0.4 \pm 0.4$ | $157.1 \pm 0.2$ | $0.8 \pm 0.2$ | $-0.7 \pm 0.1$ | $2.0 \pm 0.2$ |
| 21 October 2020 | 9143 | $88.1 \pm 0.1$ | $-0.4 \pm 0.1$ | $0.5 \pm 0.2$ | $147.9 \pm 0.1$ | $-0.8 \pm 0.1$ | $0.1 \pm 0.1$ | $1.5 \pm 0.1$ |
| 25 October 2020 | 9147 | $84.6 \pm 0.1$ | $-0.5 \pm 0.3$ | $0.9 \pm 0.4$ | $151.8 \pm 0.3$ | $-0.3 \pm 0.1$ | $-0.2 \pm 0.1$ | $0.4 \pm 0.1$ |
| 19 November 2020 | 9172 | $88.6 \pm 0.1$ | $-0.5 \pm 0.1$ | $-0.5 \pm 0.2$ | $149.8 \pm 0.1$ | $0.1 \pm 0.1$ | $-0.1 \pm 0.1$ | $0.7 \pm 0.2$ |
| 14 December 2020 | 9197 | $96.2 \pm 0.2$ | $0.1 \pm 0.2$ | $-0.5 \pm 0.4$ | $150.1 \pm 0.6$ | $1.3 \pm 0.2$ | $-0.1 \pm 0.4$ | $2.3 \pm 0.1$ |
| 18 December 2020 | 9201 | $97.4 \pm 0.1$ | $-0.1 \pm 0.1$ | $-0.8 \pm 0.6$ | $149.0 \pm 0.1$ | $0.2 \pm 0.3$ | $0.1 \pm 0.3$ | $1.4 \pm 0.4$ |
| 29 August 2021 | 9455 | $79.3 \pm 0.1$ | $0.7 \pm 0.1$ | $0.6 \pm 0.2$ | $146.8 \pm 0.1$ | $0.8 \pm 0.1$ | $1.1 \pm 0.1$ | $2.3 \pm 0.1$ |
| 7 September 2021 | 9464 | $77.7 \pm 1.1$ | $0.2 \pm 0.2$ | $0.3 \pm 2.1$ | $145.4 \pm 0.1$ | $0.4 \pm 0.5$ | $-0.4 \pm 0.1$ | $0.9 \pm 0.1$ |
| 29 October 2022 | 9881 | $75.8 \pm 0.1$ | $0.8 \pm 0.1$ | $-0.2 \pm 0.6$ | $117.8 \pm 0.6$ | $0.4 \pm 0.1$ | $-0.7 \pm 0.2$ | $1.5 \pm 0.2$ |
| 1 November 2022 | 9884 | $77.9 \pm 0.1$ | $0.7 \pm 0.1$ | $-1.1 \pm 0.4$ | $118.0 \pm 0.6$ | $1.0 \pm 0.1$ | $-0.6 \pm 0.1$ | $1.4 \pm 0.1$ |
| | | | | Mrk 335 (AFOSC) | | | | |
| 9 September 2020 | 9101 | $102.3 \pm 0.1$ | $0.7 \pm 0.1$ | $-0.4 \pm 0.2$ | $153.7 \pm 0.2$ | $0.0 \pm 0.1$ | $0.5 \pm 0.1$ | $0.8 \pm 0.1$ |
| 7 October 2020 | 9129 | $104.6 \pm 0.1$ | $-0.2 \pm 0.2$ | $0.0 \pm 0.5$ | $148.7 \pm 0.1$ | $0.6 \pm 0.3$ | $-0.5 \pm 0.4$ | $1.0 \pm 0.1$ |
| 24 November 2020 | 9177 | $102.8 \pm 0.2$ | $0.2 \pm 0.3$ | $0.0 \pm 0.6$ | $143.7 \pm 0.1$ | $0.6 \pm 0.2$ | $0.2 \pm 0.1$ | $0.9 \pm 0.1$ |
| 25 November 2020 | 9178 | $104.1 \pm 0.1$ | $-0.2 \pm 0.8$ | $-0.3 \pm 0.8$ | $145.4 \pm 0.2$ | $0.8 \pm 0.1$ | $0.3 \pm 0.4$ | $1.6 \pm 0.1$ |
| 26 November 2020 | 9179 | $103.8 \pm 0.1$ | $0.1 \pm 0.4$ | $-0.2 \pm 0.7$ | $144.6 \pm 0.2$ | $0.6 \pm 0.1$ | $0.2 \pm 0.2$ | $2.8 \pm 0.1$ |
| 28 September 2021 | 9485 | $93.9 \pm 0.1$ | $0.6 \pm 0.4$ | $-0.1 \pm 0.5$ | $140.9 \pm 0.1$ | $0.7 \pm 0.2$ | $0.2 \pm 0.3$ | $1.2 \pm 0.2$ |
| 12 October 2021 | 9499 | $92.2 \pm 0.1$ | $0.3 \pm 0.1$ | $-0.3 \pm 0.5$ | $137.2 \pm 0.1$ | $-0.3 \pm 0.1$ | $0.6 \pm 0.1$ | $0.8 \pm 0.1$ |
| 13 October 2021 | 9500 | $91.7 \pm 0.4$ | $0.5 \pm 0.1$ | $0.2 \pm 0.3$ | $138.1 \pm 0.1$ | $0.4 \pm 0.1$ | $0.7 \pm 0.1$ | $1.2 \pm 0.1$ |
| 14 October 2021 | 9501 | $92.5 \pm 0.1$ | $0.4 \pm 0.1$ | $0.1 \pm 0.6$ | $137.0 \pm 0.1$ | $0.3 \pm 0.1$ | $0.3 \pm 0.4$ | $0.9 \pm 0.1$ |
| 13 December 2021 | 9561 | $87.5 \pm 0.1$ | $0.4 \pm 0.1$ | $0.2 \pm 0.9$ | $134.8 \pm 0.3$ | $0.4 \pm 0.2$ | $0.3 \pm 0.4$ | $1.2 \pm 0.3$ |
| 11 January 2022 | 9590 | $85.2 \pm 0.1$ | $0.5 \pm 0.5$ | $-0.3 \pm 0.9$ | $128.5 \pm 0.1$ | $0.0 \pm 0.1$ | $0.7 \pm 0.1$ | $1.4 \pm 0.1$ |
| 20 August 2022 | 9811 | $77.6 \pm 0.1$ | $0.9 \pm 0.4$ | $0.2 \pm 0.6$ | $119.3 \pm 0.2$ | $0.4 \pm 0.1$ | $0.3 \pm 0.2$ | $0.8 \pm 0.1$ |
| 31 October 2022 | 9883 | $79.0 \pm 0.1$ | $0.4 \pm 0.2$ | $-0.3 \pm 0.8$ | $116.3 \pm 0.1$ | $0.9 \pm 0.1$ | $0.3 \pm 0.4$ | $1.1 \pm 0.1$ |
| | | | | Mrk 509 | | | | |
| 26 May 2020 | 8995 | $11.1 \pm 0.1$ | $0.2 \pm 0.1$ | $-0.4 \pm 0.1$ | $18.9 \pm 0.1$ | $2.3 \pm 0.1$ | $0.4 \pm 0.1$ | $0.4 \pm 0.1$ |
| 21 June 2020 | 9021 | $12.0 \pm 0.1$ | $0.2 \pm 0.1$ | $-0.1 \pm 0.1$ | $18.5 \pm 0.1$ | $3.1 \pm 0.1$ | $1.3 \pm 0.1$ | $0.6 \pm 0.1$ |
| 1 July 2020 | 9031 | $12.2 \pm 0.1$ | $0.4 \pm 0.1$ | $0.1 \pm 0.3$ | $18.1 \pm 0.1$ | $2.6 \pm 0.1$ | $1.0 \pm 0.1$ | $0.5 \pm 0.1$ |
| 23 July 2020 | 9053 | $10.7 \pm 0.9$ | $0.4 \pm 0.1$ | $0.6 \pm 0.2$ | $19.8 \pm 0.8$ | $2.6 \pm 0.1$ | $0.4 \pm 0.1$ | $0.5 \pm 0.1$ |
| 23 August 2020 | 9084 | $12.3 \pm 0.1$ | $0.9 \pm 0.1$ | $-0.6 \pm 0.2$ | $19.1 \pm 0.1$ | $2.9 \pm 0.2$ | $0.0 \pm 0.1$ | $0.5 \pm 0.1$ |
| 27 August 2020 | 9088 | $12.2 \pm 0.1$ | $0.5 \pm 0.2$ | $-0.3 \pm 0.1$ | $19.2 \pm 0.1$ | $1.9 \pm 0.2$ | $-0.7 \pm 0.1$ | $0.4 \pm 0.1$ |
| 20 September 2020 | 9112 | $13.0 \pm 0.1$ | $-0.7 \pm 0.1$ | $-0.4 \pm 0.2$ | $23.6 \pm 0.1$ | $0.1 \pm 0.1$ | $-0.5 \pm 0.1$ | $0.1 \pm 0.1$ |
| 22 September 2020 | 9114 | $12.3 \pm 0.1$ | $-0.3 \pm 0.1$ | $-1.4 \pm 0.1$ | $23.0 \pm 0.1$ | $-0.5 \pm 0.1$ | $-1.0 \pm 0.1$ | $0.2 \pm 0.1$ |
| 21 October 2020 | 9143 | $13.3 \pm 0.1$ | $1.7 \pm 0.1$ | $-0.3 \pm 0.2$ | $22.4 \pm 0.1$ | $1.8 \pm 0.1$ | $-0.9 \pm 0.1$ | $0.4 \pm 0.1$ |
| 4 August 2021 | 9430 | $12.4 \pm 0.1$ | $-0.1 \pm 0.1$ | $-1.0 \pm 0.2$ | $23.7 \pm 0.1$ | $0.0 \pm 0.1$ | $-1.4 \pm 0.1$ | $0.3 \pm 0.1$ |
| 29 August 2021 | 9455 | $12.6 \pm 0.1$ | $1.0 \pm 0.1$ | $-0.9 \pm 0.2$ | $24.6 \pm 0.1$ | $0.8 \pm 0.1$ | $-0.7 \pm 0.1$ | $0.2 \pm 0.1$ |

**Table A1.** *Cont.*

| Date | JD | $I_{cont}$ | $Q_{cont}$ | $U_{cont}$ | $I_{line}$ (Blue) $I_{line}$ (Red) | $Q_{line}$ (Blue) $Q_{line}$ (Red) | $U_{line}$ (Blue) $U_{line}$ (Red) | $I_{line}^{p}$ (Blue) $I_{line}^{p}$ (Red) |
|---|---|---|---|---|---|---|---|---|
| | | | | | Mrk 817 | | | |
| 14 December 2020 | 9197 | $24.0 \pm 0.1$ | $0.0 \pm 0.1$ | $-0.6 \pm 0.2$ | $34.7 \pm 0.1$ $24.2 \pm 0.1$ | $-0.1 \pm 0.1$ $-1.0 \pm 0.5$ | $-1.5 \pm 0.4$ $-0.7 \pm 0.7$ | $0.7 \pm 0.2$ $0.4 \pm 0.1$ |
| 18 December 2020 | 9201 | $25.0 \pm 0.1$ | $1.2 \pm 0.1$ | $0.2 \pm 0.1$ | $36.5 \pm 0.1$ $24.9 \pm 0.1$ | $-1.1 \pm 0.3$ $-2.1 \pm 0.1$ | $-0.9 \pm 0.5$ $-0.2 \pm 0.1$ | $0.7 \pm 0.1$ $0.6 \pm 0.1$ |
| 7 March 2021 | 9280 | $25.3 \pm 0.1$ | $-1.7 \pm 0.1$ | $-1.0 \pm 0.1$ | $38.9 \pm 0.2$ $25.5 \pm 0.1$ | $-2.1 \pm 0.3$ $-2.9 \pm 0.1$ | $-0.1 \pm 0.2$ $-0.6 \pm 0.1$ | $0.9 \pm 0.2$ $0.8 \pm 0.1$ |
| 8 March 2021 | 9281 | $24.9 \pm 0.1$ | $-1.9 \pm 0.2$ | $-1.1 \pm 0.1$ | $38.7 \pm 0.1$ $22.4 \pm 2.5$ | $-1.9 \pm 0.1$ $-1.7 \pm 0.2$ | $-0.5 \pm 0.1$ $-0.6 \pm 0.6$ | $0.9 \pm 0.1$ $0.4 \pm 0.1$ |
| 5 May 2021 | 9339 | $23.6 \pm 0.1$ | $-2.6 \pm 0.2$ | $-0.8 \pm 0.1$ | $38.2 \pm 0.1$ $23.5 \pm 0.1$ | $-1.1 \pm 0.1$ $-1.6 \pm 0.6$ | $-0.5 \pm 0.3$ $-2.1 \pm 0.1$ | $0.4 \pm 0.1$ $0.3 \pm 0.1$ |
| 2 July 2021 | 9397 | $22.2 \pm 0.1$ | $-2.4 \pm 0.4$ | $1.4 \pm 0.1$ | $36.7 \pm 0.8$ $23.3 \pm 0.2$ | $-1.0 \pm 0.3$ $-2.8 \pm 1.6$ | $1.7 \pm 0.4$ $2.7 \pm 0.1$ | $0.8 \pm 0.1$ $0.7 \pm 0.1$ |
| 7 July 2021 | 9402 | $21.6 \pm 0.1$ | $-1.4 \pm 0.1$ | $-0.7 \pm 0.1$ | $36.2 \pm 1.3$ $20.3 \pm 2.0$ | $-1.0 \pm 0.1$ $-1.6 \pm 0.1$ | $-0.3 \pm 0.1$ $-1.1 \pm 0.1$ | $0.5 \pm 0.1$ $0.3 \pm 0.1$ |
| 28 August 2021 | 9454 | $23.2 \pm 0.1$ | $-1.6 \pm 0.8$ | $-0.6 \pm 0.1$ | $35.4 \pm 0.1$ $22.6 \pm 0.1$ | $0.1 \pm 0.1$ $-2.0 \pm 0.1$ | $-0.2 \pm 0.3$ $-3.9 \pm 0.1$ | $0.6 \pm 0.1$ $0.8 \pm 0.2$ |

## Notes

[1] It is worth remembering that it was the approach using narrow spectral bands in the photometric observations of Seyfert galaxies that was used in the pioneering work Cherepashchuk and Lyutyi [29].

[2] Details about the characteristics of medium-band filters can be found on the https://www.sao.ru/hq/lsfvo/devices/scorpio-2/filters_eng.html (accessed on 31 December 2022).

[3] Details describing the instrument can be found on the https://www.oapd.inaf.it/sede-di-asiago/telescopes-and-instrumentations/copernico-182cm-telescope/afosc (accessed on 31 December 2022).

[4] The multiplication of SED by the filter response function is usually called "convolution" in literature yet is not equal to the real mathematical convolution.

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
