# Peer review of "Polarimetric Reverberation Mapping in Medium-Band Filters"

_universe, doi:10.3390/universe9010052_

Round 1
Reviewer 1 Report
L82-83 is "STOP" and "StoP" the same instrument? if so please unify the spelling
L91: "was achieved the scale is" ==> "was achieved. The scale is" (??)
L128-129: The text says that sigma_P/P <~0.7 means low signal-to-noise ratio. P is the polarization and sigma_P is not defined, but it would be natural to expect it to be the uncertainty of P. But if this is so the > and < signs are inverted in those two lines.
Eq 1 - the definition of I seems somewhat unusual, typically it is the sum of squared vectors from two orthogonal directions. If you sum up both 0-90 and 45-135 isn't it counting the total intensity twice?
Eq 7 is assuming a simple linear averaging of the Stokes parameters over the wavelength. I think this is too simplistic. Consider a case in which half of the band has a linear polarization with phi of 0deg and P of 1%, and the second half the same P of 1%, but orthogonal angle of 90degree. Since the different wavelengths cannot produce a constructive interference the resulting polarization should be close to 0, while the formula would give still 1%. This is particularly important because there are large swings of the polarization angle over small wavelength ranges visible in Fig 1 and 2.
Fig2 in the caption of right panels the filters from the left panels are mentioned
It is not clear if the spectropolarimetry spectra and the image polarimetry points are taken contemporaneously. If not, there can be additional effects from the AGN variability. The date of the spectropolarimetry spectra should be also mentioned in Fig 1 and2
Table 2: the actual line should be mentioned in the caption or in the table.
L197-200 - those lines are a repetition of the legend of Fig 3 and can be dropped.
Fig 3: It would make the comparisons (in particular the mentioned in the text differences) easier if Asiago and Zeiss points are put on the same panels (with different colours as they are now)
Fig 5: the caption says "polarized broad line emission" - but from the rest of the caption (I_line, I_cont) it does not seem to be polarized
L225 (and also later on): time delays with very small uncertainties are given, however the DCF in Fig 4 has very large uncertainty bars. How significant (including the trials of different time lags) is the actual delay? I am also somewhat surprised about the value of 157 days - it is longer than the individual observation periods of the source, but shorter than the gaps between those periods, hence it is strange to have a power to determine such time delay. This might further suggest this as an artifact of the analysis.
L253 while the sentence talks about time delay the units are given as "lt days" instead of "days"
L339: being in focus ==> is in focus
L405: "data with a quality better than 3" was used" - I guess you mean seeing. If so please write it clearly
Author Response
The authors sincerely thank the reviewer for the valuable comments. The text was revised and the needed clarifications are added. We also would like to mention that to make our data more available to the readers we completed Table A1 with the Stokes parameters Q and U taken from continuum and broad emission line observations. Below, we answer the reviewer’s comments.
L82-83 is "STOP" and "StoP" the same instrument? if so please unify the spelling
Corrected.
L91: "was achieved the scale is" ==> "was achieved. The scale is" (??)
Corrected.
L128-129: The text says that sigma_P/P <~0.7 means low signal-to-noise ratio. P is the polarization and sigma_P is not defined, but it would be natural to expect it to be the uncertainty of P. But if this is so the > and < signs are inverted in those two lines.
The misprint is corrected, and $\sigma_P$ definition is added.
Eq 1 - the definition of I seems somewhat unusual, typically it is the sum of squared vectors from two orthogonal directions. If you sum up both 0-90 and 45-135 isn't it counting the total intensity twice?
Indeed, the total sum of 0+90+45+135 gives doubled total intensity when one uses single birefringent prism for observations (e.g. ordinary Wollaston prism). Here we use a double Wollaston prism, so the incoming light is firstly divided into two beam for each of the two birefringent prisms, and then each of the prisms forms a double image of the incoming beam. So, in total, the four images received on the CCD frame comes from the same the incoming beam, and Eq. 1 relates to the total intensity of it.
Eq 7 is assuming a simple linear averaging of the Stokes parameters over the wavelength. I think this is too simplistic. Consider a case in which half of the band has a linear polarization with phi of 0deg and P of 1%, and the second half the same P of 1%, but orthogonal angle of 90degree. Since the different wavelengths cannot produce a constructive interference the resulting polarization should be close to 0, while the formula would give still 1%. This is particularly important because there are large swings of the polarization angle over small wavelength ranges visible in Fig 1 and 2.
Thank you for an essential comment. Indeed, the formula used in Eq. 7 does not ideally fit to estimate polarization degree value. Now we have corrected Table 1, where Eq. 7 is used to calculate Q and U parameters, and P and phi are calculated using Eq. 4 and 5. Also, we note hare and in the text that the multiplication of the spectrum by the filter response function is usually called "convolution"{} in literature yet is not equal to the real mathematical convolution. Moreover, in the same part of the text we extended the brief discussion of the expected values of polarization. The main idea is that the swing seen in the spectropolarimetry could not be resolved in polarimetry in filters, but this approach could indicate a difference in the line and continuum polarization parameters. To sum up, the text is revised according to this comment.
Fig2 in the caption of right panels the filters from the left panels are mentioned
Corrected.
It is not clear if the spectropolarimetry spectra and the image polarimetry points are taken contemporaneously. If not, there can be additional effects from the AGN variability. The date of the spectropolarimetry spectra should be also mentioned in Fig 1 and2
Corrected.
Table 2: the actual line should be mentioned in the caption or in the table.
Corrected.
L197-200 - those lines are a repetition of the legend of Fig 3 and can be dropped.
The text is corrected.
Fig 3: It would make the comparisons (in particular the mentioned in the text differences) easier if Asiago and Zeiss points are put on the same panels (with different colours as they are now)
The updated Fig. 3 is given in the text.
Fig 5: the caption says "polarized broad line emission" - but from the rest of the caption (I_line, I_cont) it does not seem to be polarized.
The misprint is corrected.
L225 (and also later on): time delays with very small uncertainties are given, however the DCF in Fig 4 has very large uncertainty bars. How significant (including the trials of different time lags) is the actual delay? I am also somewhat surprised about the value of 157 days - it is longer than the individual observation periods of the source, but shorter than the gaps between those periods, hence it is strange to have a power to determine such time delay. This might further suggest this as an artifact of the analysis.
The uncertainties given here in the text are calculated formally approximating the peak given by JAVELIN histograms with the Gaussian function and calculating the width of the profile. This comment is added. ZDCF curve was not used in these calculation and just illustrated the DCF behavior. Then, we fully agree with the reviewer’s concerns of the measured value of about 150 days. We also discuss this result in the text as unconfident.
L253 while the sentence talks about time delay the units are given as "lt days" instead of "days"
Corrected.
L339: being in focus ==> is in focus
Corrected.
L405: "data with a quality better than 3" was used" - I guess you mean seeing. If so please write it clearly
Corrected.
Reviewer 2 Report
The paper reports initial observational trials of the polarimetric reverberation mapping which may be appropriate for small telescopes.
Rather than the spectropolarimetric technique, which is more time-consuming and requires special equipment, the authors adopt a photometric polarization observation with filters that covers H\alpha line.
Though the results are still preliminary, the technique may be of use to investigate the physical dimensions of AGNs, so I would recommend the manuscript be accepted for publication after minor corrections.
I list the points I have noticed below.
(1) On page 5, lines 128-129
Please give $\sigma_P$ its definition. Also, is it correct that
"the signal-to-noise ratio of the measured polarization in AGNs was small" corresponds to $\sigma_P/P \lesssim 0.7$ and "high signal-to-noise ratio" corresponds to $\sigma_P/P>0.7$? Please check the
inequalities.
(2) On page 6, line 138, "combination of one or two"
"Combination of one" sounds odd.
(3) On page 10, in the caption of Fig.8, "Time delay analysis of the polarized"
Do you mean "non-polarized" here?
(4) For Mrk 509, R_{sc} is estimated but not R_{BLR}. Why is it?
(5) On page 13, lines 305-307, "According to.. not to be reliable."
Here the authors write that $R_{W1}:R_K=0.6:1$ and $R_{W1}\approx 1300$ lt days, then $R_K\approx 770$ lt days. I wonder how it is possible.
Author Response
The authors sincerely thank the reviewer for the valuable comments. The text was revised and the needed clarifications are added. We also would like to mention that to make our data more available to the readers we completed Table A1 with the Stokes parameters Q and U taken from continuum and broad emission line observations. Below, we answer the reviewer’s comments.
(1) On page 5, lines 128-129
Please give $\sigma_P$ its definition. Also, is it correct that "the signal-to-noise ratio of the measured polarization in AGNs was small" corresponds to $\sigma_P/P \lesssim 0.7$ and "high signal-to-noise ratio" corresponds to $\sigma_P/P>0.7$? Please check the inequalities.
The misprint is corrected, and $\sigma_P$ definition is added.
(2) On page 6, line 138, "combination of one or two"
"Combination of one" sounds odd.
Corrected.
(3) On page 10, in the caption of Fig.8, "Time delay analysis of the polarized"
Do you mean "non-polarized" here?
Corrected.
(4) For Mrk 509, R_{sc} is estimated but not R_{BLR}. Why is it?
We thank the reviewer for this comment. We have extended our analysis and included the estimation of R_{BLR}. New JAVELIN histogram for the non-polarized line flux and description are added to the text.
(5) On page 13, lines 305-307, "According to.. not to be reliable."
Here the authors write that $R_{W1}:R_K=0.6:1$ and $R_{W1}\approx 1300$ lt days, then $R_K\approx 770$ lt days. I wonder how it is possible.
There is a misprint in the relation, the true one is: $R_{K}:R_{W1}=0.6:1$. Corrected.